# Observation of a bilayer superfluid with interlayer coherence

Erik Rydow [1] ✉, Vijay Pal Singh [2], Abel Beregi [1], En Chang [1], Ludwig Mathey [3,4], Christopher J. Foot [1] & Shinichi Sunami [1] ✉

Controlling the coupling between different degrees of freedom in many-body systems is a powerful technique for engineering novel phases of matter. We create a bilayer system of two-dimensional (2D) ultracold Bose gases and demonstrate the controlled generation of bulk coherence through tunable interlayer Josephson coupling. We probe the resulting correlation properties of both phase modes of the bilayer system: the symmetric phase mode is studied via a noise-correlation method, while the antisymmetric phase fluctuations are directly captured by matter-wave interferometry. The measured correlation functions for both of these modes exhibit a crossover from short-range to quasi-long-range order above a coupling-dependent critical point, thus providing direct evidence of bilayer superfluidity mediated by interlayer coupling. We map out the phase diagram and interpret it with renormalization-group theory and Monte Carlo simulations. Additionally, we elucidate the underlying mechanism through the observation of suppressed vortex excitations in the antisymmetric mode.

Coherent Josephson tunneling between macroscopic quantum systems is an important paradigm that is the foundation for various quantum technologies[1,2]. The interplay between coupling-induced coherence and the intrinsic fluctuations of low dimensional constituent systems gives rise to a rich variety of quantum many-body phenomena[3,4]. In bilayer two-dimensional (2D) systems, this coupling can induce a transition to an interlayer superfluid state. This transition modifies the superfluid-normal transition observed in uncoupled systems, which is governed by the unbinding of vortex-antivortex pairs, known as the Berezinskii-Kosterlitz-Thouless (BKT) transition[5,6]. Such a bilayer system serves as a model with potential significance for understanding high-temperature superconductivity[7–9], including optically pumped superconductivity[10,11], twisted bilayer graphene[12,13], and dipolar particles with competing repulsive intra-plane and attractive inter-plane interactions[14,15]. Furthermore, novel phases are expected to emerge from the ordering of relative or common degrees of freedom, and there is strong interest in both the static and dynamic properties of these phases[16–24]. Several studies predict the existence of a coupling-induced superfluid phase[23,25] with others predicting a separate paired BKT superfluid phase[24,26,27], though these predictions remain largely unexplored experimentally.

Ultracold atom systems offer an exemplary platform for studying coupled many-body systems, thanks to their exquisite control over coherent quantum tunneling and the ability to directly probe many-body states; matter-wave interferometry, a key technique in cold-atom systems, provides a direct probe of relative phase fluctuations[28,29]. In addition, recent development of noise interferometry[30,31] now enables the detection of common-mode correlation properties from the density noise patterns appearing in expanded bilayer 2D systems. Although trapping of 1D and 3D quantum gases in controllable double-well potentials has been used to investigate coupled systems[32,33], the experimental realization of a tunable double-layer 2D system was not achieved before the work reported here.

We report on the creation of a highly controllable bilayer of 2D Bose gases coupled via Josephson tunneling and detailed measurements of its correlation properties using matter-wave and noise

[1]Clarendon Laboratory, University of Oxford, Oxford, United Kingdom. [2]Quantum Research Centre, Technology Innovation Institute, Abu Dhabi, UAE. [3]Zentrum für Optische Quantentechnologien und Institut für Quantenphysik, Universität Hamburg, Hamburg, Germany. [4]The Hamburg Centre for Ultrafast Imaging, Hamburg, Germany. ✉e-mail: erik.rydow@physics.ox.ac.uk; shinichi.sunami@physics.ox.ac.uk

interferometry, to probe both relative and common degrees of freedom. We fit the correlation function with algebraic and exponential models to identify the superfluid-normal transition, which manifests as a coupling-dependent crossover. This allows us to detect the emergence of a double-layer superfluid and trace the corresponding phase diagram, which agrees with renormalization group (RG) analysis of the bilayer XY model[17,23] and Monte Carlo simulations. The microscopic origin of this emergent phase is the suppression of vortex unbinding, which we confirm through direct measurements of free vortices in the relative-phase mode.

## Results

In our experimental apparatus, a cloud of $^{87}$Rb atoms is confined in a cylindrically-symmetric 2D trap formed by a box-like potential of radius 20 μm in the horizontal plane and a double-well potential in the vertical $z$ direction[34,35]. Strong vertical confinement in the double-well is created by a multiple-RF (MRF) dressing technique, as described in[29,35], while the horizontal trapping comes from the dipole force of a strong off-resonant laser beam that is spatially shaped by a digital micromirror device into a ring-shaped intensity distribution[31,36] (Fig. 1a). Atoms are loaded into the double well with equal populations at a temperature of $T = 50$ nK, set by forced evaporation. In each well the vertical trap frequency is $\omega_z/2\pi = 1.2$ kHz and the quasi-2D conditions $\hbar\omega_z > k_B T$ and $\hbar\omega_z > \mu$ are satisfied, where $\hbar$ is the reduced Planck constant, $k_B$ the Boltzmann constant and $\mu$ is the chemical potential. The characteristic dimensionless 2D interaction strength is $\tilde{g} = \sqrt{8\pi} a_s/\ell_0 = 0.08$, where $a_s$ is the s-wave scattering length and

$\ell_0 = \sqrt{\hbar/(m\omega_z)}$ is the harmonic oscillator length along $z$ for an atom of mass $m$.

The MRF-dressed double-well potential is created using RF magnetic fields with three frequency components applied to atoms in a static magnetic field gradient[37]. The separation of the two potential minima along the $z$ direction is determined by the frequency difference between the RF components. The high controllability and stability of the RF fields allow precise tuning of the inter-well distance $d$, thereby creating a bilayer system with tunable coupling strength $J$ (Fig. 1a). The interlayer coupling shifts the vortex binding-unbinding critical point, as illustrated in Fig. 1b, based on the RG theory presented in refs. 17,23 (see Supplemental Material). This emergent phenomenon affects both antisymmetric (relative) and symmetric (common) phase modes of the system, which are defined as $\theta = \phi_1 - \phi_2$ and $\varphi = \phi_1 + \phi_2$, respectively, where $\phi_i = \arg(\Psi_i)$ is the argument of the order parameter $\Psi_i$ for layer $i$ ($i = 1,2$). The relative and common modes of the system provide a natural basis for excitations in two-mode low-dimensional quantum gases[4,38], including the coupled bilayer systems[23].

We probe the spatial coherence of both phase modes using time-of-flight (TOF) expansion of the two clouds, combined with a spatially selective imaging technique along orthogonal directions, as schematically shown in Fig. 1c and d. For the relative mode, the trap is abruptly turned off, releasing the pair of 2D gases for a TOF duration of $t_{TOF}^{rel} = 17$ ms. Once released, the two clouds expand rapidly along the $z$ direction[39] and overlap, forming an interference pattern along $z$ (Fig. 1c), whose phase encodes fluctuations of the relative mode[29,40].

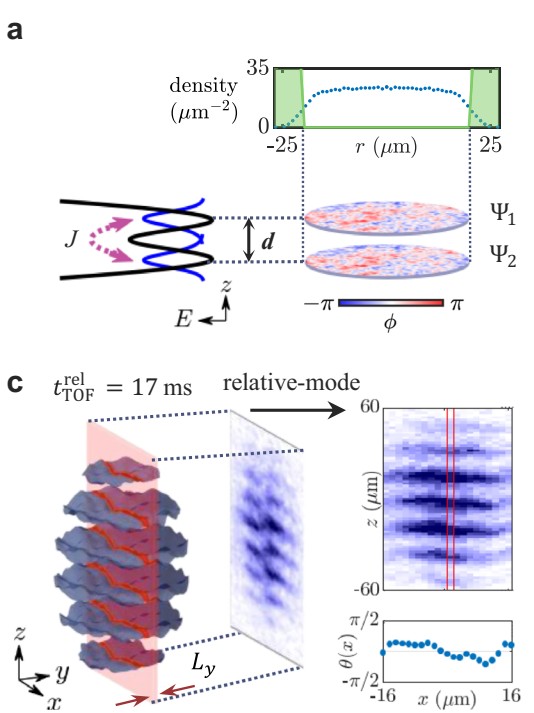

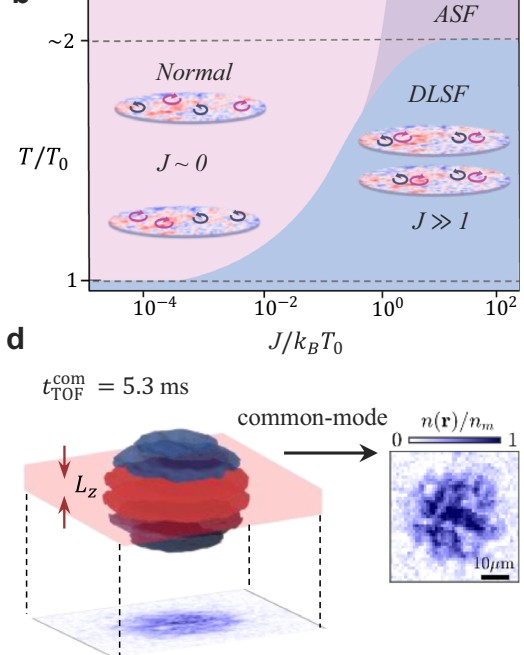

**Fig. 1 | Formation of a bilayer quasi-2D Bose gas and its characterization by matter-wave interferometry. a** We trap two near-homogeneous clouds of $^{87}$Rb atoms (represented by wave functions $\Psi_1$ and $\Psi_2$, with complex phases $\phi$) in a double-well potential, where the inter-well distance $d$ is controlled using a multiple-RF dressing technique; see text. This results in a bilayer system with a tunable interlayer coupling $J$. The top panel shows the radially averaged density profile, obtained from a single in-situ image taken along the $z$ direction. The green-shaded region indicates the box potential shape, which is created by a ring-shaped, blue-detuned laser beam. **b** Theoretical phase diagram of our coupled bilayer system based on RG analysis and Monte Carlo simulation (see Supplemental Material). Increasing the interlayer coupling $J$ increases the transition temperature $T$, towards $T/T_0 \sim 2$[23,24], where $T_0$ denotes the transition temperature for $J = 0$. Illustrations

show unbound vortex pairs in the normal phase and bound vortex pairs in the double-layer superfluid (DLSF) phase. In the anti-symmetric superfluid (ASF) phase, vortices are bound in the relative-mode but unbound in the common-mode[23]. **c** Clouds released from the trap undergo a time-of-flight (TOF) expansion for a duration of $t_{TOF}^{rel} = 17$ ms, so that they overlap producing interference fringes (blue wavy planes) encoding the local relative phase fluctuations. We capture the interference pattern by selectively imaging atoms within a thin slice of thickness $L_y = 5$ μm (shown as a red sheet; see text). The column interference profiles at different $x$ allow us to extract the local relative phase $\theta(x)$. **d** After a short TOF of $t_{TOF}^{com} = 5.3$ ms, we image the in-plane density distribution $n(\mathbf{r})$ from below using a selective imaging technique (thin repumping sheet with thickness $L_z = 5$ μm). Image on the right displays $n(\mathbf{r})/n_m$, where $n_m$ is the maximum density.

We then apply a thin sheet of laser light to optically pump atoms from the lower to the upper hyperfine level in a slice of thickness $L_y = 5\,\mu m$ along the $y$ direction (red transparent sheet in Fig. 1c) and image the repumped atoms using resonant light[34]. From the interference image we determine the local relative phase $\theta(x)$, and from a set of measurements of $\theta(x)$ we calculate the relative phase correlation function.

To probe the common mode, we use a short TOF of duration $t_{TOF}^{com} = 5.3$ ms and record the density noise patterns after expansion (see Fig. 1d). In this method, self-interference within and between the clouds transforms initial phase fluctuations into density modulations[30,41–44]. This short-TOF technique has been applied to measure phase coherence in low dimensional gases in several experiments[31,45,46], and for our density-balanced bilayer, it measures the fluctuations of the common mode, as demonstrated in ref. 31. We perform selective repumping of the atoms using a horizontal sheet of thickness $L_z = 5\,\mu m$, and image the resulting density distribution $n(x, y)$ with a high-resolution imaging system with optical axis along the $z$ direction (Fig. 1d). The selective imaging is necessary because the extent of the cloud after TOF expansion exceeds the depth of focus of the imaging system[31,47]. We explore a range of interlayer coupling strengths, from $J/h < 10^{-3}$ to $> 10$ Hz, by varying the inter-well distance $d$ between 1.7 and 5.9 μm. We cover a wide range of the phase-space density (PSD), $\mathcal{D} = n\lambda_{th}^2$, by adjusting the total atom number $N$ from $2 \times 10^4$ to $9 \times 10^4$, where $n$ is the 2D atom density in each cloud, and $\lambda_{th} = h/\sqrt{2\pi m k_B T}$ is the thermal de Broglie wavelength. For each combination of $d$ and $N$, we repeat the experiment to collect an ensemble of images using both relative and common

detection techniques. From these measurements of $\theta(x)$ and $n(x, y)$ we compute the correlation functions as described below. We average over up to 60 experimental repetitions for each set of $d$ and $\mathcal{D}$.

The real part of the two-point relative-phase correlation function is defined as $C(x, x') = \text{Re}[\langle e^{i[\theta(x)-\theta(x')]} \rangle]$, where $\theta(x)$ is the phase of the relative mode. Throughout this paper, $\langle .. \rangle$ denotes the statistical average over experimental repetitions. A value of $C(x, x') = 1$ indicates perfect coherence, while $C(x, x') = 0$ implies no coherence. In Fig. 2a, we plot $C(x, x')$ for an inter-well distance of $d = 1.7\,\mu m$ at two different phase-space densities $\mathcal{D} = 6.5$ and $3.4$. For small $\mathcal{D}$, phase coherence decays rapidly over large distances. To quantify this, we calculate the correlation function $C(\bar{x}) = \text{Re}[\langle e^{i[\theta(x)-\theta(x-\bar{x})]} \rangle_x]$ as a function of separation $\bar{x} = x - x'$, where $\langle .. \rangle_x$ denotes both the statistical average and an average over the coordinate $x$. This analysis is performed using $\theta(x)$ obtained as illustrated in Fig. 1c restricted to the central region of the cloud (see "Methods"). Figure 2b, shows that $C(\bar{x})$ decays more rapidly with distance as $\mathcal{D}$ decreases, indicating a transition from quasi-long-range order to short-range phase coherence. To identify the critical point, we fit the data with both algebraic and exponential decay models (solid and dashed lines). The reduced $\chi^2$ value ($\chi_r^2$) of the exponential fit increases significantly beyond a certain point, crossing the $\chi_r^2$ statistic for the algebraic fit. We identify this crossing as the critical value $\mathcal{D}_c$ (inset of Fig. 2b). For $d = 1.7\,\mu m$, corresponding to the coupling strength $J/h \simeq 30$ Hz, we determine $\mathcal{D}_c = 4.8(6)$ from the relative phase. This is lower than the critical value $\mathcal{D}_c(0) = 10(1)$ observed in the uncoupled system at $J/h \ll 1$ Hz.

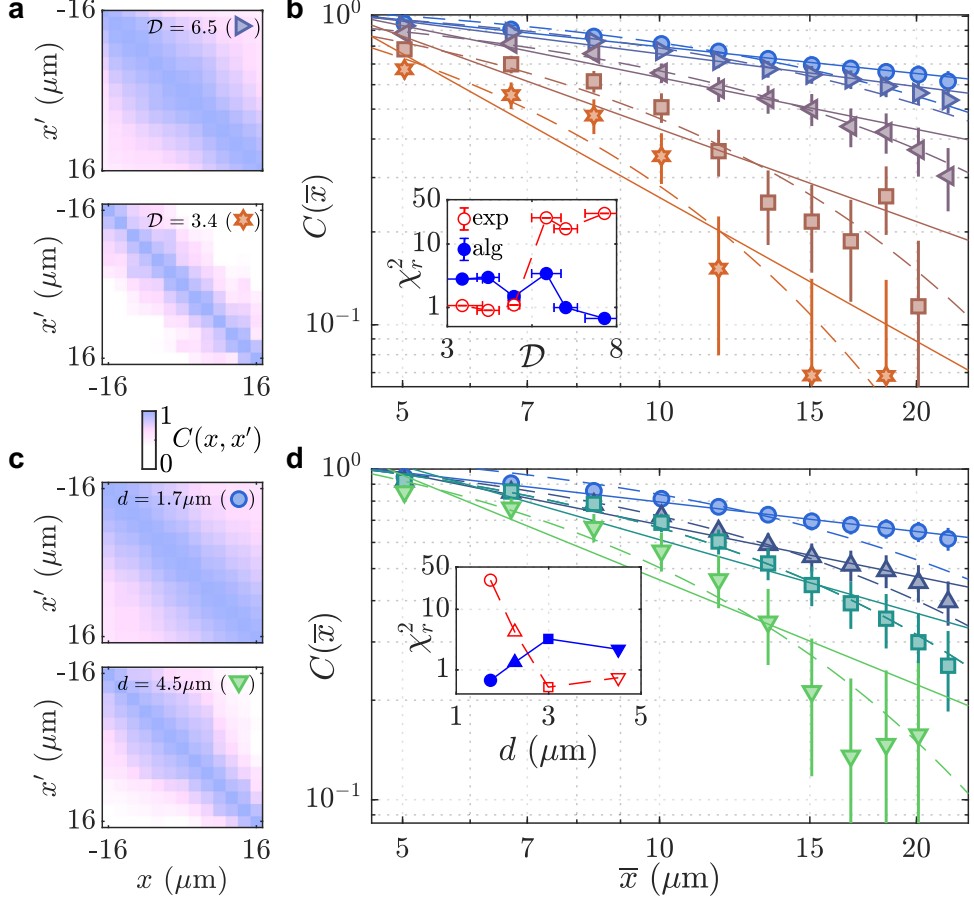

**Fig. 2 | Phase coherence of the relative mode in the coupled bilayer. a** Two-point relative-phase correlation function $C(x, x')$ is shown for phase-space densities $\mathcal{D} = 6.5$ and $3.4$, with an inter-well distance of $d = 1.7\,\mu m$. **b** Correlation function $C(\bar{x})$ plotted as a function of the distance $\bar{x} = x - x'$, measured at $d = 1.7\,\mu m$ for five different values of $\mathcal{D} = 7.6, 5.9, 5.0, 4.2$ and $3.4$ (from top to bottom). **c** $C(x, x')$ is

shown for $\mathcal{D} = 7.5$, with inter-well distances of $d = 1.7\,\mu m$ and $4.5\,\mu m$. **d**, $C(\bar{x})$ is measured at $\mathcal{D} = 7.5$ for four inter-well distances $d = 1.7, 2.3, 3.0$ and $4.5\,\mu m$ (from top to bottom). In (**b, d**), solid lines represent fits using an algebraic model, while dashed lines represent exponential model fits. Insets show $\chi_r^2$ values for the algebraic (filled symbols) and exponential (open symbols) fit models.

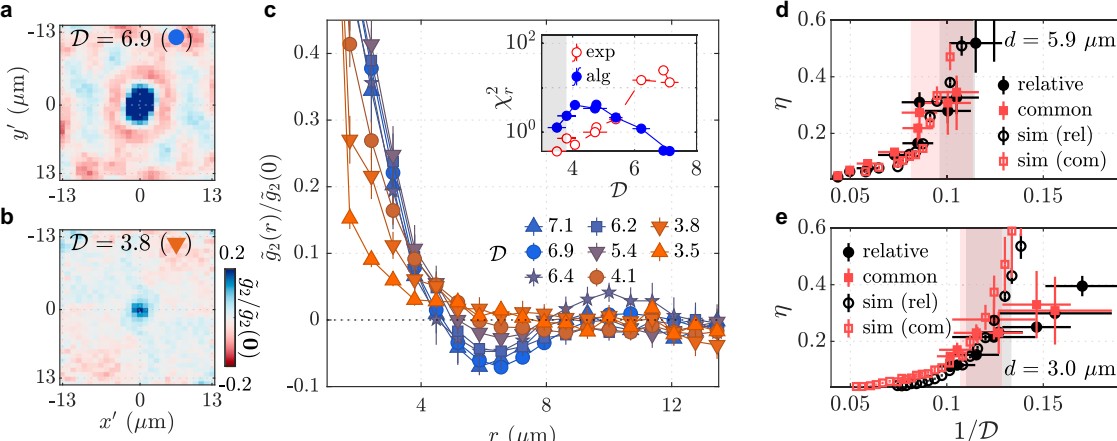

**Fig. 3 | Phase coherence of the common mode. a, b** Noise correlation functions $\tilde{g}_2(r)/\tilde{g}_2(0)$ are shown for $\mathcal{D} = 6.9$ and 3.8, with an inter-well distance $d = 1.7\,\mu m$. **c** Radially averaged noise correlation functions $\tilde{g}_2(r)$ are presented for values in the range from $\mathcal{D} = 3.5$ to 7.1, at $d = 1.7\,\mu m$, where the lines connecting the points are the guide to the eye. Inset shows the $\chi_r^2$ values for the two fit models at different $\mathcal{D}$ values (see text). Fitting is performed for $r > 2\,\mu m$ to exclude the effect of finite imaging resolution. **d, e** Measurements of the algebraic exponent $\eta$ for both the common and relative modes, along with simulation results, are shown for $d = 5.9\,\mu m$ and $3\,\mu m$. The black (red) shaded region represents the critical points and their uncertainty in the relative (common) phase, obtained from experimental data, determined by the range over which the $\chi_r^2$ values for the models cross (see Supplemental Material).

To better assess the effect of coupling on the phase coherence, we also perform measurements for varying $d$ at fixed $\mathcal{D}$. In Fig. 2c, the measurements of $C(x, x')$ at two different values of $d$ clearly indicate a fast-decaying correlation at large distance $\bar{x}$ when $d$ is increased. In Fig. 2d, the correlation functions for four distinct $d$ values show a coupling-induced crossover from algebraic to exponential phase-coherence decay. This is confirmed by fits to the two models (inset). The transition occurs around $d \simeq 2.5\,\mu m$ (or equivalently $J/h \simeq 10\,Hz$ for our system) with $\mathcal{D} \simeq 7.5$. Despite $\mathcal{D}$ being below $\mathcal{D}_c(0)$, stronger coupling suppresses phase fluctuations, enforcing algebraic order.

We now consider the information that can be deduced from the noise correlation function after free expansion $g_2(\boldsymbol{r}) = \langle \tilde{n}(\boldsymbol{r})\tilde{n}(\boldsymbol{r} - \boldsymbol{r}_0)\rangle_{\boldsymbol{r}_0}$, where $\tilde{n}(\boldsymbol{r}) = \delta n(\boldsymbol{r})/\bar{n}(\boldsymbol{r})$ is the normalized in-plane density distribution after a short TOF expansion (see Fig. 1d). Here, $\delta n(\boldsymbol{r}) = n(\boldsymbol{r}) - \bar{n}(\boldsymbol{r})$, with $\bar{n}(\boldsymbol{r}) \equiv \langle n(\boldsymbol{r})\rangle$ being the average over many experimental realizations. For bilayer systems, $g_2(\boldsymbol{r})$ the spatial auto-correlations of the density distribution after TOF encodes $\Re[\langle e^{i[\varphi(x)-\varphi(x-\bar{x})]}\rangle_x]$ the two-point correlation function of the common-mode phase in-situ[31]. Using this approach, we deduce comprehensive information about correlations of the common-mode phase from the experimental data (see "Methods"), including their functional form and values of the parameters.

Figure 3 a and b show the measurements of the density correlation function for the bilayer system at two different values of $\mathcal{D}$. At higher $\mathcal{D}$, a negative ring-like structure is visible, but this feature disappears at lower $\mathcal{D}$. This structure arises from the quasi-long-range order of the superfluid phase, which vanishes when coherence decays exponentially in the normal phase[30,31]. We theoretically calculate the noise correlation function for expanding clouds below and above the BKT transition, which we fit to our measurements to characterize the phase of the system (Fig. 3c). At PSDs much lower than the critical value $\mathcal{D} \lesssim 4$, the increase of in-situ density fluctuations affects the measurement as indicated by the gray-shaded region in the inset of Fig. 3c. By repeating this analysis for various values of $\mathcal{D}$ we determine the critical value $\mathcal{D}_c$ for our bilayer system at varying coupling strengths. Furthermore, this analysis allows us to extract the algebraic exponent $\eta$ of the superfluid phase, shown for two different inter-well distances $d = 5.9$ and $3\,\mu m$ in Fig. 3d, e. These results agree well with the measurements of the relative-phase correlations. The superfluid-normal transition occurs at a lower value of $\mathcal{D}$ when $d$ is smaller,

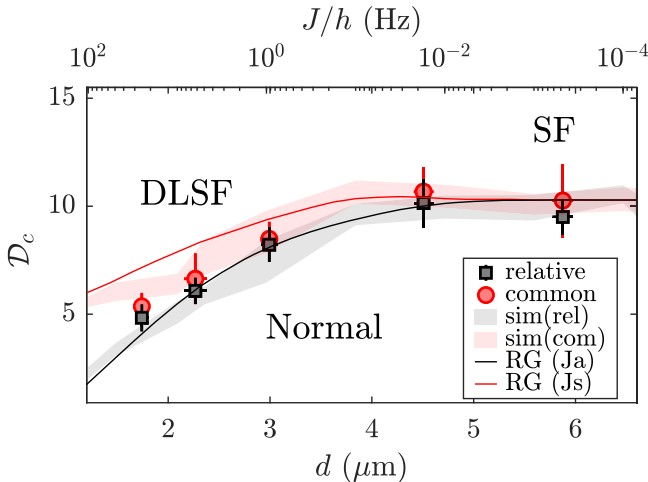

**Fig. 4 | Phase diagram of the coupled bilayer 2D Bose gas.** Measurements of the critical phase-space density $\mathcal{D}_c$ for both the relative (circles) and common (squares) modes are compared with the results from Monte Carlo simulations (filled curves). The solid lines are the predictions from the RG theory for two coupled 2D Bose gases (see Supplemental Material). The coupling strength $J$ (horizontal axis at the top) varies exponentially with the interlayer separation $d$ shown on the bottom axis (see Supplemental Material).

with $\mathcal{D}_c = 5.4(6)$ from the common phase for $d = 1.7\,\mu m$. These observations are further supported by Monte Carlo simulations, showing consistent scaling in the superfluid and crossover regimes (see Supplemental Material).

In Fig. 4, we summarize our measurements of the critical points for the relative and common modes. Within experimental error, the value of $\mathcal{D}_c$ is not strongly affected by small interlayer coupling $J/h \ll 1\,Hz$. However, $\mathcal{D}_c$ decreases monotonically with increasing coupling when $J/h \gtrsim 1\,Hz$, providing evidence for the emergence of a double-layer superfluid (DLSF) phase. These measurements agree well with the predictions of RG theory for layered 2D systems[23]. To further validate our results, we perform Monte Carlo simulations of the coupled bilayer system using experimental parameters (see Supplemental Material). From the simulations, we determine $\mathcal{D}_c(J)$ by direct

correlation analysis of the fluctuating classical fields. The simulation results for the antisymmetric mode agree closely with the respective measurements (Fig. 4). These simulations reveal that the critical points for the relative and common modes differ significantly for $d \lesssim 2\,\mu m$, indicating a strong phase-locking effect that results in an ordered relative phase while the common phase remains disordered, characteristic of a predicted anti-symmetric superfluid (ASF) phase[23]. In the range of coupling strengths that we have investigated experimentally, we have not observed the separation of the critical points for the relative and common degrees of freedom that is characteristic of a predicted ASF phase. The interesting ASF phase can be investigated, in future work, with stronger interlayer coupling by reducing the barrier height of the double-well potential (which was fixed in this work) while carefully maintaining 2D conditions for the two layers. Stronger coupling can also be engineered by other methods, such as Rabi coupling between $F = 1$ and $F = 2$ manifolds of $^{87}$Rb atoms[48].

To elucidate the microscopic origin of the DLSF phase, we analyze quantized vortex excitations which appear as sudden phase jumps in the relative-phase interference patterns (Fig. 5a). These free vortices in the relative-phase mode are only visible when they are located within the narrow region of the imaging slice (Fig. 1d), allowing the quantitative analysis of their number density from interference images[29]. In Fig. 5b, the measurements of the dimensionless vortex density $n_v \xi^2$, as

a function of $\mathcal{D}$, display exponential behaviors for all values of $d$. The healing length $\xi = 1/\sqrt{\tilde{g}n}$, determined using the 2D density $n$ and interaction $\tilde{g}$, characterizes the size of the vortex core. This exponential scaling is a hallmark of the BKT transition, consistent with previous measurements of 2D Bose gases with negligible interlayer coupling[29]. In our bilayer system, the interlayer coupling strongly suppresses vortex formation, although the scaling remains exponential. Notably, the scaling exponent increases as $d$ decreases, indicating that stronger interlayer coupling enhances vortex suppression (Fig. 5c).

The realization of bilayer 2D systems and the interferometric detection scheme demonstrated in this work provides a powerful approach for exploring novel phases in coupled systems. For instance, this platform can be utilized to study the two-step BKT transition predicted in imbalanced bilayer systems[18,19]. Moreover, the ability to tune the coupling strength, provided by MRF-potentials, makes it possible to investigate the dynamics of phenomena that were previously inaccessible, such as the Kibble-Zurek mechanism[23,49,50], universal scaling[35,50,51] in particular predictions for non-thermal fixed points in the sine-Gordon model[52,53], parametric enhancement of superfluidity[10,54], and phase-locking effect of the antisymmetric superfluid phase[23,55].

## Methods
### Monte Carlo simulation
We use classical Monte Carlo simulations to obtain the many-body thermal state of our interacting system at nonzero temperature. To perform these simulations, we discretize real space on a 2D square lattice and represent the continuous Hamiltonian using the discrete Bose-Hubbard Hamiltonian. The system consists of two subsystems (labelled $a = 1, 2$) coupled by a tunable Josephson tunneling $J$, and is described by the Hamiltonian

$$H = H_1 + H_2 + H_{12}, \tag{1}$$

with

$$H_a = -J_h \sum_{\langle ij \rangle} \left( \psi_{a,i}^* \psi_{a,j} + \psi_{a,j}^* \psi_{a,i} \right) + \frac{U}{2} \sum_i n_{a,i}^2 + \sum_i (V_i - \mu) n_{a,i} \tag{2}$$

and

$$H_{12} = -J \sum_i \left( \psi_{1,i}^* \psi_{2,i} + \psi_{2,i}^* \psi_{1,i} \right). \tag{3}$$

Here, $\langle ij \rangle$ denotes nearest neighbors, $\psi_{a,i}$ and $n_{a,i} = |\psi_{a,i}|^2$ are the complex-valued field and the density at site $i$, respectively. $V_i$ corresponds to the trapping potential at site $i$, $J_h$ is the hopping energy, and $U$ is the onsite repulsive interaction energy. We choose the simulation parameters according to the experiments. The total atom number $N$, which varies between 20,000 and 90,000, is adjusted by the chemical potential $\mu$ in the simulations. We consider a lattice system with sites $N_x \times N_y = 100 \times 100$ and use a discretization length of $l = 0.5\,\mu m$. For the continuum limit, $l$ is chosen to be smaller than or comparable to the healing length and the de Broglie wavelength[56]. The value of $U$ is determined by $U/J_h = \sqrt{32\pi} a_s/l_0 = 0.16$, based on the experimental scattering length $a_s$ and the harmonic oscillator length $l_0 = \sqrt{\hbar/(m\omega_z)}$ of the confining potential $m\omega_z^2 z^2/2$ in the transverse direction, where $m$ is the atomic mass. $J_h$ is given by $J_h = \hbar^2/(2ml^2)$, yielding $J_h/k_B = 11.16$ nK for $^{87}$Rb atoms and $l = 0.5\,\mu m$. $V_i$ is chosen such that the simulated cloud produces a homogeneous density profile with a radius of 20 μm.

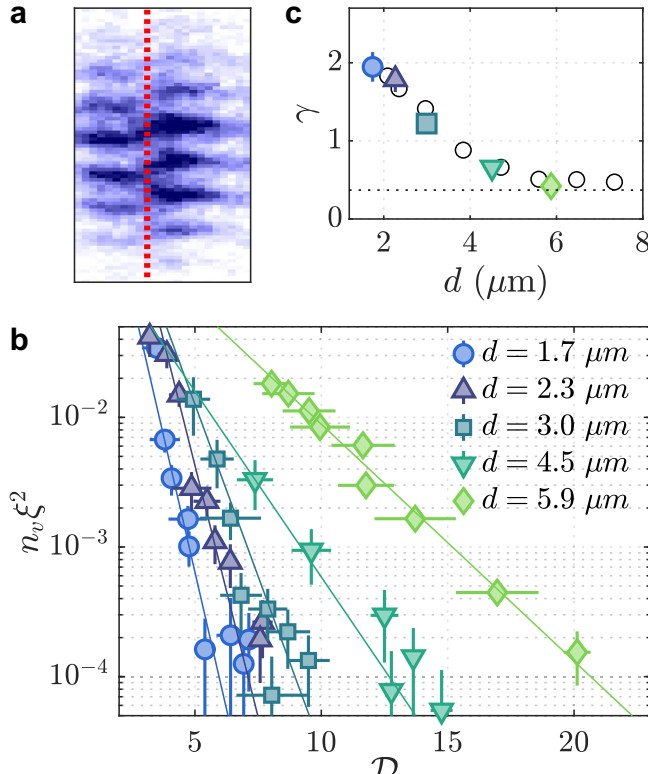

**Fig. 5 | Vortex suppression. a** Phase jumps, corresponding to vortices (as indicated by the red dashed line), emerge in the interference patterns as the system approaches the transition point. **b** Dimensionless vortex density $n_v \xi^2$ plotted on a log scale as a function of the phase-space density $\mathcal{D}$ for various values of $d$. $n_v$ is obtained by averaging over multiple experimental repetitions, with over 20 possible vortex locations sampled on each image giving a total of nearly 1000 possible locations for each datapoint, ensuring sufficient statistics for the parameter range shown. The solid lines denote exponential fits to the function $f(\mathcal{D}) = A \exp(-\gamma \mathcal{D})$, where $A$ and $\gamma$ are fitting parameters. **c** The best-fit values of the exponent $\gamma$ are shown, with the horizontal dashed line marking the value for an uncoupled system ($d = 7\,\mu m$), as reported in ref. 29. The empty circles are the results obtained from Monte Carlo simulations.

In figure legend (panel b):
$d = 1.7\ \mu m$
$d = 2.3\ \mu m$
$d = 3.0\ \mu m$
$d = 4.5\ \mu m$
$d = 5.9\ \mu m$

In the classical-field approximation, we replace the operators $\hat{\psi}$ by complex numbers $\psi$ as in Eq. (1). The initial states are generated using a grand-canonical ensemble of temperature $T$ and chemical potential $\mu$, via a classical Metropolis algorithm[57,58]. We set $T/J_h = 4.5$ and vary $\mu$ to achieve the desired $N$ for various values of $J/h$ within the range between $10^{-4}$ Hz and 100 Hz. The simulation procedure involves randomly selecting lattice sites and performing single-site updates by modifying the real and imaginary parts of the complex field, drawn from a normal distribution. The width of the distribution is adjusted such that the acceptance rate is around one half for each step. About $10^5$ steps are performed to thermalize the system. After thermalization, more than 2000 updates per site are executed to ensure that the generated states are uncorrelated. For each sample, we calculate the phases $\theta_1(x, y)$ and $\theta_2(x, y)$ of the two clouds, and use them to compute the correlation functions (Supplementary Fig. 2). We average the two-point correlation function over the initial ensemble and determine the superfluid-normal transition point for various values of $J$.

## Experimental procedure

We form the double-well potential for the dressed atoms using a combination of a static and radiofrequency (RF) magnetic fields[37,59]. The static field is a quadrupole magnetic field with cylindrical symmetry about a vertical axis, and three RF fields are applied to give a multiple-RF (MRF) double-well trap[60,61]. Control over the amplitudes and frequencies of RF components allows us to shape the potential from a single well into a double-well potential[48,60–62]. In this work, we use the combinations of RFs [7.08, 7.2, 7.32] MHz to realize well separation of $d = 5.9\,\mu m$, [7.11, 7.2, 7.29] MHz for $d = 4.5\,\mu m$, [7.14, 7.2, 7.26] MHz for $d = 3.0\,\mu m$, [7.15, 7.2, 7.25] MHz for $d = 2.3\,\mu m$ and [7.155, 7.2, 7.245] MHz for $d = 1.7\,\mu m$ (see Supplementary Fig. 1). For each set of RF frequencies, we find combinations of RF amplitudes that provide tight confinement in the vertical direction ($\omega_z/2\pi = 1.2$ kHz) for each well and produce 2D potential, with the double-well barrier height of $E_b/h \sim 4$ kHz. The large barrier height $k_B T, \mu \ll E_b$ ensures that the atoms in each plane are kinematically constrained to their respective 2D planes[63], with finite probability of hopping between the layers facilitated by the overlap of ground-state wavepackets along the $z$ direction, analogous to double-well experiments with 1D Bose gases[4,64,65].

After loading the atoms into a single-RF dressed potential and performing evaporative cooling, we transfer the atoms into the MRF-dressed potential adiabatically, by slowly introducing the other two RF signals. This can be performed with negligible heating in the system, and we further ramp up the optical potential over 3 seconds to realize a near-uniform density of atoms in the $x − y$ plane. An optical potential is created by 532 nm laser light, shaped by a spatial light modulator (digital micromirror device, DMD), to realize a box-like trap geometry (see Fig. 1a). We ensure the populations in the two wells are equal by maximizing the observed matter-wave interference contrast[34]. After equilibrating the gases further for 500 ms, the MRF-dressed potential and the optical potential are turned off, releasing the cloud into TOF expansion to observe the matter-wave interference pattern as shown in Fig. 1[29].

Finally, to probe the density distribution locally, before absorption imaging we apply a sheet of repumping light that propagates horizontally (in the $x$ direction) with thickness $L_y = 5\,\mu m$ and height much larger than the extent of the cloud of atoms[34]. All atoms are initially in a state with $F = 1$, and are then selectively pumped to $F = 2$ by the sheet of repumping light, which we image using a light resonant for the atoms in the $F = 2$ state (Fig. 1c, d). We ensure the repumping light passes through the centre of the cloud by moving the pattern along the direction parallel to the propagation of imaging light, to the position where the total absorption signal is maximum. We repeat the experiments using repumping light sheet with size covering the entire cloud, to extract the total atom number reported in the main text.

## Image analysis for relative phase detection

The analysis of the interference patterns is described in detail in refs. 29,35,66 and proceeds as follows. We first characterize the wavevector of the fringes by fitting the interference pattern with the function[40]

$$\rho_x(z) = \rho_0 \exp\left(-z^2/2\sigma^2\right)\left[1 + c_0 \cos(kz + \theta(x))\right], \qquad (4)$$

where $\rho_0, \sigma, c_0, k, \theta(x)$ are the fit parameters, as shown in Supplementary Fig. 1. We then obtain the relative phase profile $\theta(x)$ by Fourier transforming the images along the $z$ direction at each $x$ and extracting the complex argument of the Fourier coefficient corresponding to the wavevector of the fringes. The extracted phase $\theta(x)$ encodes a specific realization of the fluctuations of the in-situ local relative phase between the pair of 2D gases. From the ensemble of at least 40 images at each $d$ and $N$, we calculate the phase correlation function $C(\overline{x}) = \Re[\langle e^{i[\theta(x) - \theta(x-\overline{x})]}\rangle_x]$ where the averaging is performed over the set of images and different positions in the cloud $x$ for which $x$ and $x - \overline{x}$ are within the central 30 μm of the density distribution of the cloud. As described and confirmed experimentally in ref. 29, the long-range behavior of this function changes from algebraic scaling $\sim r^{-\eta}$ in the superfluid phase, to exponential scaling in the normal phase. We thus fit the obtained $C(\overline{x})$ at long distance $r \gtrsim 5$ μm where the effect of finite imaging resolution is negligible. From the fits with both algebraic and exponential models, we compare the $\chi_r^2$ statistics to identify the critical point $\mathcal{D}_c$ (see Fig. 2). The uncertainty of $\mathcal{D}_c$ is determined by the averaged difference of $\mathcal{D}$ to the two nearest data points.

From the interference images, taken along the $y$ direction, we detect vortices using the method described in detail in ref. 29. We look for sudden jumps of the phases within two pixel distance (3.4 μm), defined by the phase difference of $2\pi/3 < \delta\theta < 4\pi/3$. The vortex density $n_v(x)$ can be obtained by dividing the probability of finding vortices in each column of the images by the vortex detection area of a single pixel column, $\ell_p L_y = 8.4\,\mu m^2$ where $\ell_p = 1.67$ μm is the image-plane pixel size.

## Image analysis for common phase detection

As analytically studied and experimentally confirmed for bilayer 2D Bose gases in ref. 31 (and independently in ref. 67 for double-well 1D Bose gases), the spatial coherence of the common phase $\varphi = \phi_1 + \phi_2$ predominantly affects the density noise pattern observed along the double-well direction (Fig. 1d). The noise correlation function in 2D Bose gases, obtained by taking the two-point density-density correlation function after a short TOF, is expressed by the common-mode and relative-mode correlation functions $\mathcal{F}_{com}(\boldsymbol{r})^2 \simeq \langle\Psi_1^\dagger(\boldsymbol{r})\Psi_2^\dagger(\boldsymbol{r})\Psi_1(\boldsymbol{0})\Psi_2(\boldsymbol{0})\rangle/n^2$ and $\mathcal{F}_{rel}(\boldsymbol{r})^2 \simeq \langle\Psi_1^\dagger(\boldsymbol{r})\Psi_2(\boldsymbol{r})\Psi_2^\dagger(\boldsymbol{0})\Psi_1(\boldsymbol{0})\rangle/n^2$ via[31]

$$g_2(\boldsymbol{r}, t) \approx \frac{1}{(2\pi)^2}\int d^2\boldsymbol{q}\int d^2\boldsymbol{R} \cos(\boldsymbol{q}\cdot\boldsymbol{r})\cos(\boldsymbol{q}\cdot\boldsymbol{R})$$
$$\times \frac{\mathcal{F}_{com}(\boldsymbol{q}_t)^2\mathcal{F}_{com}(\boldsymbol{R})^2}{\mathcal{F}_{com}(\boldsymbol{R} - \boldsymbol{q}_t)\mathcal{F}_{com}(\boldsymbol{R} + \boldsymbol{q}_t)}\mathcal{F}_{rel}(\boldsymbol{q}_t)^2, \qquad (5)$$

where $\boldsymbol{q}_t = \hbar q t/m$ and $t$ is the time-of-flight duration. The common-mode fluctuations are primarily responsible for the spatial structure of the self-interference patterns and thus the oscillatory behavior of $g_2$, while relative-mode correlations are only relevant in the normal phase, where $g_2$ displays short-ranged exponential decay (Fig. 3c).

The analysis of the density noise patterns, as shown in Fig. 3, proceeds as follows, as described in ref. 31. From at least 20 experimental images taken from below as in Fig. 1d for each experimental parameter value, we first normalize the images by the average density distribution for each dataset. We then obtain autocorrelations from the density patterns in the images within a region of interest (ROI) which captures the central part of the cloud. This results in a collection of correlation functions on a 2D grid, scaled by the squared mean

density $n_0^2 = \langle \hat{n}(\boldsymbol{r}, t) \rangle^2 = \langle \hat{\Psi}(\boldsymbol{r}, t)^\dagger \hat{\Psi}(\boldsymbol{r}, t) \rangle^2$, where $\hat{\Psi}(\boldsymbol{r}, t)$ is the bosonic field operator after the expansion, which corresponds to[30]

$$\frac{\langle \hat{n}(\boldsymbol{r}, t)\hat{n}(\boldsymbol{0}, t) \rangle}{n_0^2} = g_2(\boldsymbol{r}, t) + \frac{\delta(\boldsymbol{r})}{n_0}, \quad (6)$$

where the second term is the shot-noise term with zero mean, such that

$$g_2(\boldsymbol{r}, t) = \frac{\langle \hat{\Psi}^\dagger(\boldsymbol{r}, t)\hat{\Psi}(\boldsymbol{r}, t)\hat{\Psi}^\dagger(\boldsymbol{0}, t)\hat{\Psi}(\boldsymbol{0}, t) \rangle}{\langle \hat{n}(\boldsymbol{r}, t) \rangle \langle \hat{n}(\boldsymbol{0}, t) \rangle}, \quad (7)$$

is identified by averaging over experimental repetitions. Some of the extracted $g_2$, such as Fig. 3a, exhibit anisotropy of the ring structure which arises because of finite sampling used in our measurements but this does not affect the radially averaged $g_2$ functions used for the quantitative analysis such as the fitting procedure described in this section. We have confirmed this by taking a larger dataset for a few chosen parameters and then randomly selecting a subset, some of which contain anisotropic rings, and finding that this resampling analysis yields the same radially averaged $g_2$ functions.

The quantitative analysis of the measured $g_2$ follows the procedure of ref. 31; we fit the experimental data with a model based on Eq. (5), where the model is constructed from the theoretical forms that describe both $\mathcal{F}_{com}$ and $\mathcal{F}_{rel}$, namely, the algebraic falloff in the superfluid regime and the exponential decay in the normal regime, for both common and relative phase modes[23,30]. More concretely, we programmed a numerical routine to compute Eq. (5) and performed curve fitting via nonlinear least squares and fit results of a few representative datasets are shown in Supplementary Fig. 4.

### Renormalization-group theory

The analytical prediction for the phase diagram in Fig. 4 is based on the renormalization-group equations for coupled 2D superfluids in refs. 17,23. The equations relate the effective system parameters at varying length scales $l$, and provide a universal description of the DLSF phase and its transitions. The coupled equations are expressed in terms of, the temperature energy scale $T'$, the interlayer coupling $J_\perp$, the stiffness of symmetric and antisymmetric phase fluctuations, $J_{s/a} = J \pm J_{int}$, the single vortex fugacity $A_1 \sim Je^{-J}$, and corresponding fugacities for symmetric and antisymmetric vortex pairs $A_s$ and $A_a$, and is[23]

$$
\begin{aligned}
\frac{dJ_\perp}{dl} &= \left(2 - \frac{T'}{2\pi J_a}\right)J_\perp, \\
\frac{dA_s}{dl} &= \left(2 - 2\pi\frac{J_s}{T'}\right)A_s + \alpha_3 \frac{A_1^2(J_a - J_s)}{2T'^2}, \\
\frac{dA_a}{dl} &= \left(2 - 2\pi\frac{J_a}{T'}\right)A_a + \alpha_3 \frac{A_1^2(J_s - J_a)}{2T'^2}, \\
\frac{dA_1}{dl} &= \left(2 - \frac{\pi(J_s + J_a)}{2T'} + \alpha_3 \frac{A_sJ_s + A_aJ_a}{T'^2}\right)A_1, \\
\frac{dJ_a}{dl} &= \alpha_2\left(\frac{J_\perp^2}{4\pi^4 J_a} - 4\frac{A_a^2}{T'^4}J_a^3 - \frac{A_1^2}{2T'^4}(J_s + J_a)J_a^2\right), \\
\frac{dJ_s}{dl} &= -\alpha_2\left(2\frac{A_s^2}{T'^4}J_s^2 + \frac{A_1^2}{4T'^4}(J_s + J_a)J_s\right)2J_s,
\end{aligned}
\quad (8)
$$

where $\alpha_2$ and $\alpha_3$ are dimensionless non-universal constants. We identified the crossover for common and relative modes shown in Fig. 4, from the behavior of $J_s$ and $J_a$, respectively, after integrating Eqs. (8) for $\Delta l$, as we vary the $\mathcal{D}$: the transition is labelled at the $\mathcal{D}$ where the dimensionless stiffnesses $\tilde{J}_{s/a} = J_{s/a}\pi/T'$ suddenly drop below a certain value, which we chose to be $10^{-1}$ for this work, where the changes jumps by orders of magnitude at the transition under RG flow (Supplementary Fig. 3). We used Bayesian optimization to identify the non-universal RG parameter values for our system, reported in Supplementary Fig. 3 caption, where the cost function is defined as

the $\chi^2$ distance between the RG phase diagram to the Monte Carlo simulation results.

### Estimation of Josephson coupling $J$

We estimate the interlayer coupling strength $J$ by numerically solving for the ground and first excited states in our trap using the imaginary time evolution of 3D Gross-Pitaevskii equation[68]. We deduce the Josephson plasma energy in the two-mode model following the method of the improved two-mode model in ref. 69. For our system the relation between well separation and Josephson coupling energy follows

$$J/h = 2437 e^{(-b \cdot d)} \, \text{Hz}, \quad (9)$$

where $b = 2.63 \times 10^6 \, \text{m}^{-1}$ and $d$ is the well separation.

## Data availability
Data supporting this study are openly available from Zenodo at ref. 70.

## Code availability
The simulation codes supporting this study are openly available from Zenodo at ref. 70.

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

## Acknowledgements

This work was supported by the EPSRC Grant Reference EP/X024601/1, and E.R. and A.B. thank the EPSRC for doctoral training funding. L.M. acknowledges support by the Deutsche Forschungsgemeinschaft (DFG, German Research Foundation), namely the Cluster of Excellence 'Advanced Imaging of Matter' (EXC 2056), Project No. 390715994. The project is co-financed by ERDF of the European Union and by 'Fonds of the Hamburg Ministry of Science, Research, Equalities and Districts (BWFGB)'.

## Author contributions

S.S. designed and E.R., A.B., E.C., and S.S. performed the experiments. V.S. and L.M. developed numerical models and contributed to the interpretation of the data. E.R., V.S., L.M., and S.S. implemented the renormalization-group model used in this work. E.R., V.S., and S.S. wrote the manuscript. L.M., C.J.F., and S.S. supervised the project. All authors contributed to the discussion and interpretation of our results.

## Competing interests

The authors declare no competing interests.
