## [Transparent Peer Review file · Nature Communications]

Observation of a Bilayer Superfluid with Interlayer Coherence

Corresponding Author: Mr Erik Rydow

Version 0:

Reviewer comments:

Reviewer #1

(Remarks to the Author)

In this work, the authors perform an experimental analysis of bilayer superfluidity by directly probing the coherence of the two-layered atomic cloud via interferometry. They observe the bilayer superfluid phase transition, by exploring the space parameters of the interlayer coupling. The interpretation of the experimental results is supported by renormalization group (RG) analysis of the bilayer XY model, and Monte Carlo (MC) simulations.

The results are expressed in a clear way and are pertinent to the scope of the journal. However, there are some points that can be formulated more clearly. In particular, the experimental considerations could be decoupled from the theory in order to better clarify the quantities derived from the experiments without direct reference to the theoretical model. Specifically

1. In order to make the article more accessible, the discussion of the common-mode phase measurement (noise interferometry, at page 1 line 31) would benefit from a brief additional explanation, of its working principle, or a link to the Methods, since it is less well-known compared to matter-wave interferometry. Moreover, how is $C(x,x')$ explicitly extracted from the experimental data, without referring to quantum field theory or the presence of two coupled order parameters? It will be useful to give some insights without deferring to Methods. The averaging performed for the computation of $C(\bar{x})$ seems different in page 9 line 605 with respect to page 4, line 146. It would be more clear to utilize the subindex x in both expressions.

2. Further commentary on the validity of the two order parameter model for this system will be best introduced before utilizing the concepts of this theoretical picture (at page 2, line 90). It would be advisable to corroborate this assumption by commenting on previous experimental studies on bilayers that utilize the same approach.

Moreover, a brief elucidation on the following issues would be advisable:

1. Are the RG and MC results reported in the phase diagram compatible with those presented in Ref. [20]?

2. In the discussion of the RG equations (M8), it would be better to define the non-universal coefficients α_2 and α_3 , that are currently not defined, and they are used in Fig. S3.

Reviewer #2

(Remarks to the Author)

Reviewer #3

(Remarks to the Author)

The authors report experiments and theoretical studies on the phenomenon of bilayer superfluidity. They consider two 2D cold gases of bosons with an intralayer tunnel coupling, which they tune by varying the distance between the two atomic planes. They determine the common and relative mode coherence of this system for different phase-space densities and varying the tunnel coupling. The data are compared with numerical simulations and an RG theory. Finally they briefly study the vortex density in the system.

This study is original and represents a breakthrough in the experimental study of bilayer systems. Even though the focus is on ultracold atoms, it is of broad interest for condensed matter physics. However, I have some major issues with the presentation of this work that should be addressed by the authors.

- 1) Generally speaking, while it is clear that this work is strongly related to recent studies on 2D gases by the same experimental team, I find that some explanations are missing in this work. The article should be self-consistent without the need to read the authors's previous references at a first level.
- 2) In this sense, I find the discussion of the common-mode coherence unclear. It is explained first that the common mode phase is $\phi = \phi_1 + \phi_2$ (page 2) and suddenly on page 3, we jump to noise correlation measurements without any explanation of the relationship between ϕ and g_2 .
- 3) The authors should clarify the definition of g_2 which, if I understand correctly, is related to the the total cloud density after expansion.
- 4) In Fig 3, there is no clear explanation of how to determine the η coefficient from the decay of g_2 . Could we see the fit that allows the authors to determine the exponents of the algebraic/exponential decay ?
- 5) Still in Fig. 3, why did the authors choose $d = 1.8 \mu\text{m}$ for Fig3c, while the values of η is given in Figs3de for other values of d ?
- 6) line 161 : it is confusing to give the value of D_c from the common phase measurement at this point in the text. Perhaps the authors could make the comparison elsewhere, for instance when discussing Fig4.
- 7) In Fig 2b, I do not think that it is a good choice to have points (stars) where we only see the top of the error bar and not the data point itself. The authors should change the scale or, if these points are not significant because they are below their detection threshold, they the should not show them.
- 8) I find the last part about the vortex density measurements interesting and with a clearly convincing signal but, I'm surprised that there is no theoretical study on this part. It seems to me that the authors have the theoretical tools to also study this vortex density and this is missing in the current version of the manuscript.
- 9) Regarding the introduction of the article, I would say that a discussion of bilayer dipolar gases is missing. I am not a specialist enough to give precise suggestions to the authors, but it seems to me that there is a substantial literature on bilayer superfluidity between two clouds coupled by magnetic dipole-dipole interactions. It would be interesting to briefly discuss the similarities and differences between the two setups so as to make larger the impact of this work.

In addition, I think that the article Hadzibabic et al. Nature 2006 about the BKT transition, which uses the same method to determine the relative phase mode should be cited.

Reviewer #4

(Remarks to the Author)

In this paper, the authors study the experimental observation of a bilayer superfluid of two-dimensional (2D) Bose gases with interlayer coherence. They observe evidence of bilayer superfluidity by observing a crossover from short-range to quasi-long-range phase coherence with increasing the interlayer coupling strength. This is further supported by the suppression of free vortices in the superfluid regime, which is the key feature of the Berezinskii-Kosterlitz-Thouless (BKT) transition in 2D Bose gases.

This is an interesting paper that provides the first experimental realization of a tunable bilayer 2D Bose gas with interlayer coherence, a major step in cold atom research. As the author introduced, it is an analogue of the high-temperature superconductivity in condensed matter physics.

Moreover, prior theoretical studies [ref. 19, 20] predicted paired BKT superfluidity in bilayer systems, but experimental confirmation has been lacking until this work.

The work presents multiple experimental techniques (matter-wave interferometry, noise correlation) to identify the BKT critical point, and the results agree well with each other. The experimental data is quite clear and well-presented. The authors also provide theoretical studies (Monte Carlo simulation and renormalization group theory) to support their observation.

Overall, the experiments are excellent and timely study that presents a major advance in bilayer superfluidity and ultracold atom physics. The manuscript is well written and organized to present the bilayer superfluidity. Some points need to be

addressed, but this work may be worth publication in Nature Communication. Below are the comments to be addressed for the publication.

The coherent tunnelling between two layers has significant importance in condensed matter physics, as seen in the high T_c superconductivity. More recently, the twisted bilayer graphene (TWG) emerged, opening a new paradigm in condensed matter systems. The recent studies of TWG could be cited and discussed in the introduction to strengthen the connection between this work and the broader context of bilayer quantum materials. Also, importantly, the author should include related works on the cold atom. [M. Gall et al., Nature (London) 589, 40 (2021) and Z. Meng et al., Nature (London) 615, 231 (2023)].

I'm a bit confused about the word "measure the symmetric phase mode of the condensate", which appeared in the abstract and the main text, line 91. The experiments, however, measure the density and its noise correlations. Of course, the correlation implies the phase information, but the experiment is not measuring the phase $\theta_1 + \theta_2$. After reading the paper I could understand what it means, but it needs to be rephrased.

In Fig.1, the author provides two different methods that read relative phase information and density fluctuations from the phase fluctuations of the global phase. Fig.1d, there is a clear density depletion core at the edge, which might suggest the vortex. Could the author simultaneously measure both images? Make a partial imaging with a short TOF and then take a second image to read out the relative phase. This could be helpful to distinguish the DLSF and ASF.

In Fig.2, the author displays a power-law decay constant η in the relative phase correlation function. This constant can be related to the superfluid density, according to ref 36 and eqn. 47. Could the authors provide an explicit comparison between the experimentally extracted superfluid density and the theoretical/numerical predictions in the BKT regime?

In the noise correlation function, the author measured the density profile for a fixed hold time of $t=5.3$ ms. Could the author show a scaling behavior with different TOF like shown in the previous studies [ref. 39, 40].

In Fig.3a, the noise correlation function displays an anisotropy. Interestingly, at high temperatures, data shows anisotropy with its axis rotated almost 90 degrees. Can the authors explain the observed anisotropy rotation in Fig. 3a?

Lines 192-193: the author claims that the experimental data in Fig. 3c is well agreed with the numerical simulation. To improve clarity, I suggest overlaying a few representative experimental curves with the numerical simulation in Fig. 3c, while moving additional data to Supplementary information.

Lines 193-196: I guess the D_c is obtained when the χ^2 for different fit functions (the algebraic and exponential functions) agree. However, the χ^2 for the algebraic function is non-monotonic and reaches its maximum around $D \sim 4$. A clear explanation and justification are required to claim the phase space density D_c in the noise correlation measurements.

Lines 219-221 and Fig. 4: The experimental data for the antisymmetric phase mode shows a very good agreement with the theory. However, the critical points obtained from the noise correlation data are not as good as the phase correlation data. Could the authors clarify why the critical points from the noise correlation data deviate from those of the phase correlation measurements? Is this due to different sensitivity to fluctuations or another systematic effect?

Lines 242-243: If the vortices in different layers are locked to each other, one would get a clear interference pattern without dislocation. I guess the experiment does not appear to enter this regime, but as shown in the theory curve of Fig. 4, there is a small window for the anti-symmetric superfluid phase. The author should declare the ASF phase has not been reached to claim the vortex suppression. Also, it would be better to discuss the limitations and potential ways to access the ASF phase. Additionally, I recommend citing Z. Hadzibabic et al., Nature (London) 441, 1118 (2006).

It would be better to elaborate more details of the outlooks. Could the authors elaborate on what specific aspects of universal scaling are currently inaccessible and how future experiments might address these limitations? Furthermore, how do these results relate to recent developments in non-thermal fixed points?

Version 1:

Reviewer comments:

Reviewer #1

(Remarks to the Author)

The authors addressed all our comments satisfactorily, and we recommend publication in Nature Communications.

Optional: Maybe the authors could be a bit more explicit in the introduction about the paired BKT phase predicted in Ref. [24] but not "directly compatible" with their experimental findings.

Reviewer #2

(Remarks to the Author)

Reviewer #3

(Remarks to the Author)

I thank the authors for their response, which I find largely satisfactory.

However, I am still concerned by their answer to point 2. I asked for clarification on the relationship between ϕ and g^2 , but I cannot find an answer to this question on page 5 as the authors stated.

Secondly, having read the new version of the article, I have identified a further issue. In the caption of Fig. 5, it is said that the results are obtained by taking "an average over multiple experimental repetitions". I would like the authors to provide a typical number of repetitions. The plotted curve (similar to Ref. 28) shows algebraic scaling over almost three decades. Assuming that for the largest vortex density, the number of detected vortices is of order 1, the authors must be measuring an average vortex number below 1%, which is quite challenging (I have neglected the ξ vs. PSD variation, which I guess is quite OK for $d = 1.7 \mu\text{m}$). Therefore, giving the typical number of repetitions more precisely seems important to me (this information is also not present in Ref. 28).

Reviewer #4

(Remarks to the Author)

I would like to thank the authors for their extensive response to my comments and also those of other referees. The authors have adequately addressed my questions and comments, clarifying some crucial issues. As a minor comment, some references are missing DOIs [Refs. 12,13,14,15,52,53], but this can be addressed during the editing process. Overall, I am satisfied with all changes made and am happy to recommend this work for publication in Nature Communications.

Reviewer #1 (Remarks to the Author):

In this work, the authors perform an experimental analysis of bilayer superfluidity by directly probing the coherence of the two-layered atomic cloud via interferometry. They observe the bilayer superfluid phase transition, by exploring the space parameters of the interlayer coupling. The interpretation of the experimental results is supported by renormalization group (RG) analysis of the bilayer XY model, and Monte Carlo (MC) simulations.

The results are expressed in a clear way and are pertinent to the scope of the journal. However, there are some points that can be formulated more clearly. In particular, the experimental considerations could be decoupled from the theory in order to better clarify the quantities derived from the experiments without direct reference to the theoretical model. Specifically

We thank the referee for the nice overall evaluation. We address the remaining comments below.

1. In order to make the article more accessible, the discussion of the common-mode phase measurement (noise interferometry, at page 1 line 31) would benefit from a brief additional explanation, of its working principle, or a link to the Methods, since it is less well-known compared to matter-wave interferometry. Moreover, how is $C(x,x')$ explicitly extracted from the experimental data, without referring to quantum field theory or the presence of two coupled order parameters? It will be useful to give some insights without deferring to Methods. The averaging performed for the computation of $C(\bar{x})$ seems different in page 9 line 605 with respect to page 4, line 146. It would be more clear to utilize the subindex x in both expressions.

The authors thank the referee for this suggestion to improve the clarity of the manuscript. We have added a brief description on page 1, as suggested. We have further updated Ref. [31] which is now published [Sunami et al., Phys. Rev. Lett. 134, 183407 (2025)]. In addition, we also added a sentence explaining the high-level working principle of noise-interferometry on page 5, and expanded the methods section, including the addition of Fig.S4 to illustrate the fitting used to analyze the density-noise correlation functions. We have modified the text to make explicit that to calculate $C(x, x')$ we use $\text{Re}\langle e^{i[\theta(x)-\theta(x')]} \rangle$ and the 1D phase profiles from the interference fringes (as in Fig.1 C), We thank the referee for highlighting the inconsistent notation and we now utilize the subindex x in both expressions.

2. Further commentary on the validity of the two order parameter model for this system will be best introduced before utilizing the concepts of this theoretical picture (at page 2, line 90). It would be advisable to corroborate this assumption by commenting on previous experimental studies on bilayers that utilize the same approach.

We thank the referee for highlighting this point. We have added a short discussion clarifying the validity of the two order parameter model on page 3, referencing previous experimental and theoretical studies utilizing this approach.

Moreover, a brief elucidation on the following issues would be advisable:

1. Are the RG and MC results reported in the phase diagram compatible with those presented in Ref. [20]?

Our results are not directly compatible with Bighin et al. [24] (previously cited as [20]), however our phase diagram is consistent with earlier results for the same Hamiltonian [Mathey et al., Eur. Phys. Lett. 81, 10008 (2007), Cazalilla et al., Phys. Rev. A 75, 051603 (2007)] as well as recent work by Xiao et al. (arXiv:2504.01461) in response to the results in [24], these indicate that there appear to be finer points left to be clarified in the work by Bighin et al. The active discourse and differing phase diagrams for the system we realize experimentally highlights the complications in theoretical analysis which arise in non-trivial many body scenarios. Our experimental and supporting theoretical work presents a timely contribution to this active debate.

To reflect the state of research regarding the paired superfluid phase in [23, 25] we have rephrased the introduction, to separate works which predict a phase-diagram with a paired BKT phase, as in [24] from the work which predicts a double-layer superfluid phase [24, 26]. We also added a reference to the work by Xiao et al. [27].

2. In the discussion of the RG equations (M8), it would be better to define the non-universal coefficients α_2 and α_3 , that are currently not defined, and they are used in Fig. S3.

We appreciate the referee for highlighting this and now introduce α_2 and α_3 in the methods section, along with the other parameters in the RG equations.

Reviewer #2 (Remarks to the Author):

Reviewer #3 (Remarks to the Author):

The authors report experiments and theoretical studies on the phenomenon of bilayer superfluidity. They consider two 2D cold gases of bosons with an intralayer tunnel coupling, which they tune by varying the distance between the two atomic planes. They determine the common and relative mode coherence of this system for different phase-space densities and varying the tunnel coupling. The data are compared with numerical simulations and an RG theory. Finally they briefly study the

vortex density in the system.

This study is original and represents a breakthrough in the experimental study of bilayer systems. Even though the focus is on ultracold atoms, it is of broad interest for condensed matter physics. However, I have some major issues with the presentation of this work that should be addressed by the authors.

We appreciate the referee's understanding of the potential significance of this technique and our result. The responses to the particular issues raised are provided below.

1) Generally speaking, while it is clear that this work is strongly related to recent studies on 2D gases by the same experimental team, I find that some explanations are missing in this work. The article should be self-consistent without the need to read the authors's previous references at a first level.

We thank the referee for highlighting this and have made several changes to make the manuscript more self-contained. These include clarifying how we calculate the relative phase correlations from our experimental data on page 4, with reference to Fig.1d. We also added text on page 5 to explain the principle behind the density-noise technique and expanded the methods section to include motivation as to why density distributions after TOF can be used to probe the coherence of the common phase. Fig. S4 has been added to the Supplementary Material to illustrate the fitting procedure used when analyzing images of the density-noise patterns; we refer to Fig. S4 in the expanded methods section. We finally note that our previous work Ref. [31] has now been published [Sunami et al., Phys. Rev. Lett. 134, 183407 (2025)].

2) In this sense, I find the discussion of the common-mode coherence unclear. It is explained first that the common mode phase is $\phi = \phi_1 + \phi_2$ (page 2) and suddenly on page 3, we jump to noise correlation measurements without any explanation of the relationship between ϕ and g_2 .

We appreciate the referee for pointing this out, we have now revised the manuscript to explicitly state the dependence of g_2 on the common mode coherence in situ, on page 5 where we introduce the density noise correlations.

3) The authors should clarify the definition of g_2 which, if I understand correctly, is related to the total cloud density after expansion.

The referee's understanding is correct. To avoid confusion with g_2 possibly denoting the in-situ density distribution we have added a clarification to the definition of g_2 on page 4, explicitly stating that it is related to the density distribution after expansion.

4) In Fig 3, there is no clear explanation of how to determine the eta coefficient from the decay of g2. Could we see the fit that allows the authors to determine the exponents of the algebraic/exponential decay ?

We perform non-linear fitting, by evaluating g_2 from M5, using the known functional forms of algebraic and exponential decay with free parameters as outlined in Methods. From this fitting we extract the associated eta parameter. Fig. S4, showing these fits, has been added to supplemental material which we refer to in the methods section. Intuitively, the anticorrelation dip evident in the high-PSD dataset is the consequence of quasi-long-range coherence, where the relative magnitude of this dip is related to the eta parameter.

5) Still in Fig. 3, why did the authors choose $d=1.8\mu\text{m}$ for Fig3c, while the values of eta is given in Figs3de for other values of d?

We thank the referee for this remark and like to clarify that different panels serve different purposes. In panels (a, b, c) we show the smallest interwell distance example that we realized with our bilayer system, and note that g_2 functions for decoupled ($J \ll 1$) systems are plotted already in our previous work [31]. Panels d and e are chosen for two different interwell distances that lie in the intermediate and rightmost regime of the phase diagram, where the latter distance corresponds to the scenario where Josephson coupling has no effect and then we compare this with the case where a sizeable coupling effect is present as evident by the shift in the critical point. So overall, the presented results span the entire interwell distance range explored in the experiments. Thus there is a smooth lead into the phase diagram as a function of separation in Fig. 4.

With this in mind, the interwell distances in panels (d, e) were chosen to keep the x-axes range the same on both panels to aid comparison between them. While the closest separation dataset shown in Fig3c the transition occurs at $1/D \sim 0.2$, the experimental data shown in Fig.3c lies in the range from $1/D \sim 0.05$ to ~ 0.15 .

6) line 161 : it is confusing to give the value of D_c from the common phase measurement at this point in the text. Perhaps the authors could make the comparison elsewhere, for instance when discussing Fig4.

We have implemented this suggestion, and we have moved where the value of D_c at the smallest separation is stated.

7) In Fig 2b, I do not think that it is a good choice to have points (stars) where we only see the top of the error bar and not the data point itself. The authors should change the scale or, if these points are not significant because they are below their detection threshold, they should not show them.

We appreciate the referee's point and following the suggestion we introduced the criterion $\text{std}(C)/C < 1$, to not include points which are much less relevant than the other datapoints in determining the correlation behavior. We slightly expanded the y axis range in Fig.2b to show two additional points which are above this threshold.

8) I find the last part about the vortex density measurements interesting and with a clearly convincing signal but, I'm surprised that there is no theoretical study on this part. It seems to me that the authors have the theoretical tools to also study this vortex density and this is missing in the current version of the manuscript.

The authors thank the referee for bringing up the possibility for greater theoretical comparisons regarding the vortex suppression. The suppression we see is consistent with what we observe in our Monte-Carlo simulation as shown in Fig.5c (open markers are for Monte-Carlo simulations). Our other theoretical tool, the renormalization analysis, predicts bulk properties by iteratively integrating over microscopic fluctuations, and thus it does not readily provide predictions regarding concrete values for the microscopic properties such as the vortex density. We are aware that there are other theoretical techniques to interpret the vortex suppression observed but we considered such analysis to be beyond the scope of this work.

9) Regarding the introduction of the article, I would say that a discussion of bilayer dipolar gases is missing. I am not a specialist enough to give precise suggestions to the authors, but it seems to me that there is a substantial literature on bilayer superfluidity between two clouds coupled by magnetic dipole-dipole interactions. It would be interesting to briefly discuss the similarities and differences between the two setups so as to make larger the impact of this work.

We greatly appreciate the referee's suggestion and have added a mention of bilayer superfluidity of dipolar gases to the introduction with brief discussion of the differences to the situation studied in our work.

In addition, I think that the article Hadzibabic et al. Nature 2006 about the BKT transition, which uses the same method to determine the relative phase mode should be cited.

This was an oversight on our part, we now include a reference to this seminal article on using matter-wave interferometry to identify the BKT transition.

Reviewer #4 (Remarks to the Author):

In this paper, the authors study the experimental observation of a bilayer superfluid of two-dimensional (2D) Bose gases with interlayer coherence. They observe evidence of bilayer

superfluidity by observing a crossover from short-range to quasi-long-range phase coherence with increasing the interlayer coupling strength. This is further supported by the suppression of free vortices in the superfluid regime, which is the key feature of the Berezinskii-Kosterlitz-Thouless (BKT) transition in 2D Bose gases.

This is an interesting paper that provides the first experimental realization of a tunable bilayer 2D Bose gas with interlayer coherence, a major step in cold atom research. As the author introduced, it is an analogue of the high-temperature superconductivity in condensed matter physics. Moreover, prior theoretical studies [ref. 19, 20] predicted paired BKT superfluidity in bilayer systems, but experimental confirmation has been lacking until this work.

The work presents multiple experimental techniques (matter-wave interferometry, noise correlation) to identify the BKT critical point, and the results agree well with each other. The experimental data is quite clear and well-presented. The authors also provide theoretical studies (Monte Carlo simulation and renormalization group theory) to support their observation.

Overall, the experiments are excellent and timely study that presents a major advance in bilayer superfluidity and ultracold atom physics. The manuscript is well written and organized to present the bilayer superfluidity. Some points need to be addressed, but this work may be worth publication in Nature Communication. Below are the comments to be addressed for the publication.

We thank the referee for the nice evaluation of our work. Below we address the remarks:

The coherent tunnelling between two layers has significant importance in condensed matter physics, as seen in the high T_c superconductivity. More recently, the twisted bilayer graphene (TWG) emerged, opening a new paradigm in condensed matter systems. The recent studies of TWG could be cited and discussed in the introduction to strengthen the connection between this work and the broader context of bilayer quantum materials. Also, importantly, the author should include related works on the cold atom. [M. Gall et al., Nature (London) 589, 40 (2021) and Z. Meng et al., Nature (London) 615, 231 (2023)].

The authors thank the referee for highlighting this connection. We now mention the relevance of coherent tunneling to TWG superconductivity in the introduction, including references to the suggested studies of TWG using cold atoms the referee suggested.

I'm a bit confused about the word "measure the symmetric phase mode of the condensate", which appeared in the abstract and the main text, line 91. The experiments, however, measure the density and its noise correlations. Of course, the correlation implies the phase information, but the experiment is not measuring the phase $\theta_1 + \theta_2$. After reading the paper I could

understand what it means, but it needs to be rephrased.

To address this point we have rephrased the abstract and the main text to clarify that we measure correlation properties that arise from the symmetric phase mode, not the phase of this mode directly.

In Fig. 1, the author provides two different methods that read relative phase information and density fluctuations from the phase fluctuations of the global phase. Fig. 1d, there is a clear density depletion core at the edge, which might suggest the vortex. Could the author simultaneously measure both images? Make a partial imaging with a short TOF and then take a second image to read out the relative phase. This could be helpful to distinguish the DLSF and ASF.

This is an insightful suggestion by the referee which would, in theory, be possible with an improved imaging system. The heating and disturbance caused by the initial imaging could be minimized by using less powerful repumping light so only a fraction of atoms in the sheet interact with the light and are imaged. Our current apparatus is limited by only having one digital micromirror device for the selective repumping, however, we will look into this idea in future work. A related method is the partial release of atoms from RF-dressed potential [Prufer et al., Phys. Rev. Lett., 133, 250403 (2024)]; this alternative and potentially powerful method allows the simultaneous readout of both common and relative phases.

In Fig. 2, the author displays a power-law decay constant η in the relative phase correlation function. This constant can be related to the superfluid density, according to ref 36 and eqn. 47. Could the authors provide an explicit comparison between the experimentally extracted superfluid density and the theoretical/numerical predictions in the BKT regime?

We appreciate the referee's point, our results qualitatively agree with the theoretical relation of the scaling of superfluid stiffness in finite sized systems. This scaling can be seen from the values of the decay coefficient η ; the experimental values are shown alongside numerically calculated values in Fig. 3d, Fig. 3e and there is agreement. The linear relationship between η and T given by Eq 47 in ref 36 indeed agrees with the linear scaling of η against $1/D$ in the superfluid regime as seen in Fig. 3d, Fig. 3e. This linear relationship was also observed in the uncoupled case, as reported in Fig. 4 of Ref. [31] in accordance with the well-known Nelson-Kosterlitz scaling.

In the noise correlation function, the author measured the density profile for a fixed hold time of $t=5.3$ ms. Could the author show a scaling behavior with different TOF like shown in the previous studies [ref. 39, 40].

This is a highly relevant point raised by the referee, however this has been included in another paper that we have submitted (Fig. 2, Ref. [31]); that figure shows scaling for different ToF as in the previous work in the comment.

In Fig.3a, the noise correlation function displays an anisotropy. Interestingly, at high temperatures, data shows anisotropy with its axis rotated almost 90 degrees. Can the authors explain the observed anisotropy rotation in Fig. 3a?

We thank the referee for highlighting the potential systematic features of the g_2 patterns in Fig.3a, and Fig.3b; the referee's comment prompted further investigation. For the different separations studied anisotropy in g_2 is not a general feature observed in either the high or low PSD regimes. This observation is in agreement with our recent work [Sunami et al., Phys. Rev. Lett. 134, 183407 (2025)].

To verify that the anisotropy in g_2 seen in Fig.3a is not an inherent systematic feature of the system studied, we have taken an additional data (in response to this comment), with the number of experimental repeats nearly factor of two larger than the dataset for Fig. 3a. This larger dataset allows random variations in the observed density arising because of finite sample size to be investigated by bootstrapped resampling, i.e. by randomly selecting 20 images out of the larger set, to reproduce the sample size used to find g_2 in Fig.3a. While slightly noisier due to differing condition of the apparatus, the resulting g_2 functions often show anisotropy in a random direction, as shown in Response Fig.1 a-d below, similar to Fig. 3a. Despite their different appearances, after performing radial averaging, the deduced radially averaged g_2 function - which is the quantity of interest used for our quantitative analysis - agrees with that of the entire dataset within error bars.

Response Fig.1: Variation in g_2 functions for 1.4 μm separation when computed from samples of 20 randomly chosen images out of 37. In panel d the radially averaged g_2 functions are shown, along with the radially averaged g_2 computed from all 37 images, shown as a black line with the uncertainty as a gray shaded region, the samples in panels a-b, all give rise to radially averaged g_2 functions.

This analysis suggests that the anisotropy in g_2 , seen in Fig.3a, and to a lesser extent in Fig.3b, is not a property of (tunnel-coupled) bilayer 2D Bose gases but it arises from finite sample number when plotted in a way shown in Fig. 3a. This artifact has no impact on the radially averaged functions that are used for quantitative analysis in the paper. As such, this does not systematically

affect our analysis. We have added a remark to this effect in the Methods section under *Image analysis for common phase detection underlining this.*

Lines 192-193: the author claims that the experimental data in Fig. 3c is well agreed with the numerical simulation. To improve clarity, I suggest overlaying a few representative experimental curves with the numerical simulation in Fig. 3c, while moving additional data to Supplementary information.

To implement this suggestion we have added Fig. S4 (in Suppl. Mat.) to show the procedure described on lines 192-193 we have added figure S4 to supplemental material.

Lines 193-196: I guess the D_c is obtained when the χ^2 for different fit functions (the algebraic and exponential functions) agree. However, the χ^2 for the algebraic function is non-monotonic and reaches its maximum around $D \sim 4$. A clear explanation and justification are required to claim the phase space density D_c in the noise correlation measurements.

We appreciate the referee for highlighting this, and we have added Fig. S4 to the supplementary material showing the best fits obtained with models with an exponentially or algebraically decaying functional form of the in-situ phase correlation function. These fits show a crossover of the functional form, in agreement with the crossing of chi-squared values shown in the inset on Fig.3c. At phase-space densities lower than the crossover, the exponential model fits better than the algebraic one and the anticorrelation dip disappears.

For even lower PSDs deep in the normal regime, the g_2 function decays to 0 at a few μm distance, effectively reducing the reliable regime for curve fitting. Furthermore, this is the regime where the increased in-situ density fluctuations affect the g_2 function at short distances i.e. for length scales less than a few μm (see also Fig.8 of PRA 89, 053612 (2014), cited as [30]). The combined effect makes the fits less stable, and we generally observe the fluctuation of χ^2 values deep in the normal regime (as seen also in Fig.3 c and f inset of Ref. [31]).

Thus, we believe this non-monotonicity is not a general feature expected from the model, but rather it is instead an artefact arising from the experimental probe. We emphasize that the parameter regime where this non-monotonicity occurs lies outside of the relevant region for our analysis and interpretation of phase diagram, and as such, does not affect our conclusion. We have added this clarification to the main text and drawn a gray shaded region for $D < 4$ in Fig. 3(c) inset to indicate the region where there is reduced reliability of the χ^2 values, as discussed above.

Lines 219-221 and Fig. 4: The experimental data for the antisymmetric phase mode shows a very good agreement with the theory. However, the critical points obtained from the noise correlation data are not as good as the phase correlation data. Could the authors clarify why the critical points from the noise correlation data deviate from those of the phase correlation measurements? Is this

due to different sensitivity to fluctuations or another systematic effect?

This is an important remark on the comparison between the RG analysis and the experiments/numerics. In Fig. 4, the experimental results and the numerical simulations demonstrate good agreement, and the RG results capture the general features of the measured phase diagram well. As the referee points out however, the common mode results for the RG are slightly shifted upwards from experimental results and the numerical simulations.

The RG analysis we have undertaken in this work, using the coupled equations from Ref. [23], is perturbative in the vortex fugacities and tunneling energy. The good agreement between the RG predictions and the experiment for relative modes, and the slight deviations between the RG and experiment for the common mode (while exhibiting the same qualitative features) may indicate that potential improvements should be made to the RG equations by incorporating higher-order terms. This may be carried out in future theoretical work.

Lines 242-243: If the vortices in different layers are locked to each other, one would get a clear interference pattern without dislocation. I guess the experiment does not appear to enter this regime, but as shown in the theory curve of Fig. 4, there is a small window for the anti-symmetric superfluid phase. The author should declare the ASF phase has not been reached to claim the vortex suppression. Also, it would be better to discuss the limitations and potential ways to access the ASF phase. Additionally, I recommend citing Z. Hadzibabic et al., Nature (London) 441, 1118 (2006).

We now clarify towards the end of page 5 that we do not observe the ASF phase, as the experimental data for the range of couplings investigated do not show a complete suppression of relative vortices or separation of critical points for relative and common modes. At the suggestion of the referee a short discussion of potential ways to reach the ASF phase has been included at the end of page 5 and start of page 6.

We now also cite the 2006 work by Z. Hadzibabic et al.

It would be better to elaborate more details of the outlooks. Could the authors elaborate on what specific aspects of universal scaling are currently inaccessible and how future experiments might address these limitations? Furthermore, how do these results relate to recent developments in non-thermal fixed points?

We appreciate the referee's point that our comment on universal scaling in the Outlook didn't incorporate recent work on the sine-Gordon model, which is naturally realised in tunnel-coupled bilayers and is therefore very relevant. Experimental results on universal scaling in the 2D sine-Gordon model would be of interest given recent theoretical developments, which we now mention.

Additional Changes:

We fixed inconsistencies in the notation used in the discussion of the hopping terms in Eqs. (M2) and (M3).

We explicitly divide J by h in (M9) so the dimensions of J are consistent within the paper.

We have addressed inconsistencies in our rounding of the distance for the bilayers with the closest separation, the fringe spacing (as seen in S1) gives $1.74\mu\text{m}$, this should be rounded to $1.7\mu\text{m}$ not $1.8\mu\text{m}$ as it was in some places.

Changed inconsistent hyphenation of “interlayer”.

We corrected the y axis label for Fig.3c. It is the same quantity as in Fig.3a, Fig.3b.

We made the cropping in Fig.3a and Fig.3b consistent.

We reduced the number of dashed lines in the insets showing Chi-squared in Fig.3 c, and Fig.2b, Fig2d.

Reviewer #1 (Remarks to the Author):

The authors addressed all our comments satisfactorily, and we recommend publication in Nature Communications.

Optional: Maybe the authors could be a bit more explicit in the introduction about the paired BKT phase predicted in Ref. [24] but not "directly compatible" with their experimental findings.

The authors appreciate this suggestion and have incorporated it to make it clearer that the findings in Ref. [24] concern a phase which is "separate" from the coupling-induced superfluid phase observed in this work.

Reviewer #2 (Remarks to the Author):

Reviewer #3 (Remarks to the Author):

I thank the authors for their response, which I find largely satisfactory.

However, I am still concerned by their answer to point 2. I asked for clarification on the relationship between ϕ and g_2 , but I cannot find an answer to this question on page 5 as the authors stated.

On page 5 lines 220-220 of the diff, we explain the relationship between g_2 and the common phase coherence in words. To increase clarity, we changed the manuscript on in this area, starting with "For bilayer systems, ", to explicitly reference the symbols g_2 and the common mode phase correlation function.

Secondly, having read the new version of the article, I have identified a further issue. In the caption of Fig. 5, it is said that the results are obtained by taking "an average over multiple experimental repetitions". I would like the authors to provide a typical number of repetitions. The plotted curve (similar to Ref. 28) shows algebraic scaling over almost three decades. Assuming that for the largest vortex density, the number of detected vortices is of order 1, the authors must be measuring an average vortex number below 1%, which is quite challenging (I have neglected the ξ vs. PSD variation, which I guess is quite OK for $d = 1.7 \mu\text{m}$). Therefore, giving the typical number of repetitions more precisely seems important to me (this information is also not present in Ref. 28).

We now provide the number of possible vortex locations in our dataset in the caption of Fig. 5, this is the number of images (>40) multiplied by the number of columns in an image of an interference pattern (~20): this makes the total statistics for vortex detection to be close to 1000, such that vortex number below 1% can be identified with sufficient confidence.

Reviewer #4 (Remarks to the Author):

I would like to thank the authors for their extensive response to my comments and also those of other referees. The authors have adequately addressed my questions and comments, clarifying some crucial issues. As a minor comment, some references are missing DOIs [Refs. 12,13,14,15,52,53], but this can be addressed during the editing process. Overall, I am satisfied with all changes made and am happy to recommend this work for publication in Nature Communications.

We are happy that the referee considers their point addressed and we greatly appreciate pointing out the missing DOIs in the references. We have added these and updated references for preprints which have since been published.